# Psychological Factors That Suppress Help-Seeking among Middle-Aged and Older Adults Living Alone

**DOI:** 10.3390/ijerph191710620

**Published:** 2022-08-25

**Authors:** Yoh Murayama, Sachiko Yamazaki, Masami Hasebe, Tomoya Takahashi, Jun Yamaguchi, Erika Kobayashi

**Affiliations:** 1Research Team for Social Participation and Community Health, Tokyo Metropolitan Institute of Gerontology, Tokyo 173-0015, Japan; 2Department of Human Studies, Bunkyo Gakuin University, Saitama-ken 356-8533, Japan; 3Department of Human Welfare, Seigakuin University, Saitama-ken 362-8585, Japan

**Keywords:** help-seeking, middle-aged and older adults, future time perspective, resignation to future, living alone

## Abstract

Help-seeking among destitute adults has not been adequately investigated. Therefore, this study clarifies the mechanisms that suppress help-seeking in middle-aged and older adults living alone. Data were collected from 1274 individuals (aged 50–79 years) who were living alone, using a survey that measured future time perspective, barriers to help-seeking, help-seeking intentions, and current and childhood economic statuses. Men living alone experienced lower help-seeking intention than women, were more likely to try to solve problems by themselves, and experienced greater distrust in others. No sex differences were observed in “future anxiety” and “resignation to the future.” Poor economic status was associated with high “resignation to the future,” “future anxiety,” and “distrust of others” for both sexes. “Resignation to the future” was particularly higher among men with a poorer current economic status, which suppressed help-seeking. Abandoning hope for the future, which is characteristic of middle-aged and older men living alone, may inhibit help-seeking behavior.

## 1. Introduction

According to the World Social Report by the United Nations Department of Economic and Social Affairs [1], income disparities and inequality are rising in two-thirds of the countries. Addressing these disparities has become a matter of global concern and requires immediate attention. Moreover, there is a pressing need to establish psychological support for people from low-income and economically marginalized backgrounds [2].

As of July 2019, approximately 1.63 million households in Japan were receiving public assistance. Of these, 820,000 individuals were older adults living alone on their own, and this number continues to rise [3]. Middle-aged and older adult men who live alone are at a particularly high risk of social isolation [4]; therefore, measures to prevent isolation among them are urgently needed [5]. Economic status is often associated with social relationships among older adults living alone [6]. Furthermore, studies have also suggested a relationship between subjective and objective measures of economic status and mental health among older adults [6,7,8]. For instance, Murayama, Hasebe, Nishi, and Fujiwara [7] found that economic status mediates the relationship between mutual aid or social support and mental health among older adults; that is, the lower the economic status, the more effective social support is in helping older adults maintain their mental health. However, the use of public services, such as community healthcare and preventive care, is low among middle-aged and older men [9]. Studies have also found that men are less likely than women to seek help for medical problems, substance abuse, and psychological distress [10,11,12,13].

The influence of traditional gender roles, which demand achievement and independence from men and dependence from women, has been identified as one of the reasons for the gender differences observed in help-seeking behavior [14,15]. Previous studies show that men are less likely to seek help to prevent perceiving a threat to the norms of masculinity [16,17]. In Japan, where traditional gender roles (“men at work, women at home”) are highly prevalent, men are expected to gain employment and sustain the family [18]. Among Japanese men, stigma is associated with unfavorable employment status and low income; in contrast, among Japanese women, stigma is associated with the types of occupations women are engaged in [19]. Thus, for middle-aged and older Japanese men, since experiencing economic difficulties is at odds with traditional norms of masculinity, negative attitudes toward help-seeking are expected in such economically unfavorable situations.

Failure to recognize the need for help, despite objectively requiring it, can be another barrier to help-seeking and tends to be more common among men living alone [20,21,22,23]. Kono, Tadaka, Okamoto, Kunii, and Yamamoto-Mitani [21] interviewed older men living alone and found that the participants had trouble perceiving their own unhealthy lifestyle habits as problems; the study emphasized the need to encourage older men to engage with their problems proactively. Murayama, Yamazaki, Hasebe, Takahashi, and Kobayashi [22] interviewed 83 single older men who had experienced hardships such as isolation or poverty during middle or older adulthood. They discovered that these men “were not thinking about the future despite facing severe difficulties in everyday life.”

The difficulty that middle-aged and older men who live alone experience in recognizing their need for help suggests that they are not thinking enough about the future. An individual’s perception of the future, referred to as future time perspective (FTP), has been shown to be correlated with well-being [24] and life satisfaction [25] among middle-aged and older adults. FTP is also strongly correlated with planning for life after retirement; that is, individuals who are high on FTP tend to be more involved in planning for post-retirement life [26,27]. In other words, it is expected that severe economic problems can inhibit an individual’s hope for their future and suppress their help-seeking behavior.

However, women are seen as more proactive help-seekers because dependence on others has been normalized as a “feminine” gender role [15]. Additionally, a previous study also showed that women use more coping strategies than men do [28,29]. In other words, it is possible that women are more proactive problem-solvers than men are and are therefore more open to the idea of seeking help. In light of these findings, this study aims to understand the mechanisms that suppress help-seeking behavior in middle-aged and older men and women who live alone. Specifically, for men, we hypothesize that due to poorer economic status, barriers to seeking help increase, and a feeling of resignation to the future becomes stronger, thereby resulting in suppression of help-seeking behavior (see hypothesis model in Figure 1). However, in the case of women, we hypothesize that a feeling of resignation to the future will have no impact on their help-seeking behavior. Given that previous studies have found that economic conditions during childhood later influence mental and physical health and social relationships [30,31], we also consider the effects of both current and childhood economic status. Through this study, we hope to clarify the relationship between gender, economic status, and help-seeking behavior among middle-aged and older adults living alone.

## 2. Materials and Methods

### 2.1. Participants

Using a random sampling method, 4000 residents were selected from 247,445 residents between the ages of 50 and 79 who were listed in the Basic Resident Register of Tokyo’s Ward A (as of January 2020) and who lived alone (response rate: 45.7% (*n* = 1829)); a postal survey was conducted in December 2020. The response rates from single-person households, urban areas, and working generations were relatively low [32], which reflects the individual characteristics in this study. We excluded data from respondents who lived with another person (*n* = 272), had left their responses incomplete, or had missed answering questions that were essential to the study. Consequently, data from 1274 participants (658 men and 616 women) were included in the analysis. This study was approved by the Research Ethics Board of the Tokyo Metropolitan Institute of Gerontology (approval number: 23; approved on 4 August 2020). The participants received a written description of the research and instructions for completing all questionnaires. We assumed that informed consent had been provided if the completed questionnaire was returned by the individual.

### 2.2. Measures

#### 2.2.1. Future Time Perspective

We used the Future Time Perspective Scale (FTP [23]), which measured perceived FTP among middle-aged and older adults living alone. It comprised two factors: one factor representing resignation to and renunciation of the future (hereafter, “resignation to the future”), and another representing anxiety related to the future (hereafter, “future anxiety”). To establish the content validity of the scale, we formally discussed the contents of the scale with experts in the fields of social welfare and clinical psychology. One additional item (“I feel bleak when I think about what comes next”) was added. The revised version of the scale, therefore, included eight items that were each rated on a four-point scale from 4 = “agree” to 1 = “disagree.”

#### 2.2.2. Psychological Barriers to Help-Seeking

Based on the Barriers to Help-Seeking Scale [33], translated from English into Japanese, and the results of a preliminary survey with 1200 participants between the ages of 50 and 79 [23], we created 10 questions to determine potential items for measuring psychological barriers to seeking help. In this study, participants were asked “What do you think about asking someone (e.g., a relative, friend, or acquaintance) for help?” on a five-point scale (5 = “strongly agree” to 1 = “strongly disagree”). We used all 10 questions to measure psychological barriers to help-seeking.

#### 2.2.3. Help-Seeking Intention

We developed the Japanese version [34] of the General Help-Seeking Questionnaire (GHSQ) [35] for middle-aged and older adults living alone. The GHSQ evaluates how likely respondents are to consider seeking help from different helpers chosen by the researcher. For this study, participants were asked to imagine a time when they were experiencing anxiety or facing a problem in their life that they could not solve on their own and to rate their likelihood of reaching out to each helper on a five-point scale (5 = “strongly agree” to 1 = “strongly disagree”). If the respondent did not have such a person in their life, they were asked to not provide any rating. The following helper categories were selected based on the results of a preliminary survey [23] as well as the attributes of the participants of this study (middle-aged and older adults living alone): (1) intimate partner, (2) friend, (3) colleague from work (including a former workplace), (4) parent, (5) sibling, (6) son/daughter, and (7) neighbor. The total score for each item was calculated and used as the help-seeking intention score. The score range is 0–35 points; the higher the score, the greater the intention of help-seeking.

#### 2.2.4. Economic Status

Participants were asked to evaluate their current financial circumstances on a five-point scale ranging from 1 = “extremely difficult” to 5 = “extremely comfortable.” To evaluate their childhood economic status, participants were asked to respond to the question, “How do you feel your family’s financial circumstances were compared to other families around you when you were 15 years old?” on a five-point scale ranging from 1 = “extremely difficult” to 5 = “extremely comfortable.”

### 2.3. Analysis

We performed exploratory factor analysis (principal factor method, promax rotation) for FTP and barriers to help-seeking and then performed a multigroup confirmatory factor analysis for men and women to calculate the measurement invariance of the factor structure obtained. Items with a factor loading of <0.40 and those that yielded high factor loadings in multiple factors were excluded. Next, we performed a *t*-test using the total score for each factor as the dependent variable and sex as the independent variable. Lastly, to examine the relationship between each measured variable, we conducted multigroup structural equation modeling (Maximum Likelihood Estimation Method) for men and women. Analyses were performed using SPSS for Windows version 24.0 (IBM Corp., Armonk, NY, USA) and AMOS for Windows version 24.0 (IBM Corp., Armonk, NY, USA). The statistical significance level was set to 5%.

## 3. Results

### 3.1. Participant Characteristics and Interaction Effects

Table 1 presents data regarding the basic attributes of the participants. A *t*-test was used to examine the gender differences. The current economic status score (t_(1272)_ = 4.01, *p* < 0.01, d = 0.23) was significantly higher for women (M = 3.01, SD = 0.93) than men (M = 2.79, SD = 1.00), and the score of childhood economic status (t_(1272)_ = 4.95, *p* < 0.01) was also significantly higher for women (M = 3.04, SD = 0.95) than men (M = 2.77, SD = 0.97, d = 0.28). A weak significant positive correlation was observed between current and childhood socioeconomic statuses (r = 0.18, *p* < 0.01).

### 3.2. Scale Structure

We conducted an exploratory factor analysis (principal factor method, promax rotation) for all eight items of FTP. One item that showed high factor loadings on multiple factors (item 3) was excluded, and the remaining seven items were reanalyzed. Consistent with Murayama et al. [23], two factors were ultimately extracted (Table 2). The confirmatory factor analysis (maximum likelihood ratio), conducted separately for men and women, resulted in the following values: Comparative Fit Index (CFI) = 0.974, Adjusted Goodness of Fit Index (AGFI) = 0.935, Root Mean Square Error of Approximation (RMSEA) = 0.081 for men, and CFI = 0.974, AGFI = 0.935, RMSEA = 0.081 for women. Although the RMSEA values were slightly above the acceptable range (0.05–0.08 [36]), we judged these to be acceptable when considered alongside the CFI and AGFI values in the case of both men and women. Then, we performed multigroup structural equation modeling for men and women. The model’s goodness-of-fit was as follows: χ^2^ = 65.757 (df = 26), *p* = 0.001, CFI = 0.988, AGFI = 0.970, RMSEA = 0.035, Akaike’s Information Criterion (AIC) = 125.757 (Model 1). Next, we analyzed the model without equality constraints for all other paths. The model’s goodness-of-fit was as follows: χ^2^ = 66.368 (df = 31), *p* = 0.001, CFI = 0.990, AGFI = 0.974, RMSEA = 0.030, AIC = 116.368 (Model 2). The goodness of fit for both models was sufficient to allow for the adoption of these models, but we selected Model 2 because of its higher CFI and AGFI and lower AIC and RMSEA values. Two factors were appropriate for men and women. Factor 1 (“future anxiety”) included three items, while Factor 2 (“resignation to the future”) included four items. A sufficiently high internal consistency coefficient for men and women was obtained for each factor (α = 0.70–0.90).

Additionally, we conducted an exploratory factor analysis for all 10 items in the Barriers to Help-Seeking Scale. One item that had a low factor loading (item 9) and two items that had high factor loadings on multiple factors (items 2 and 3) were excluded. The remaining seven items were reanalyzed. Two factors were extracted (Table 3). The confirmatory factor analysis (maximum likelihood ratio), conducted separately for men and women, indicated the following values: CFI = 0.988, AGFI = 0.965, RMSEA = 0.055 for men, and CFI = 0.994, AGFI = 0.971, RMSEA = 0.045 for women. We judged these values to be acceptable in the case of both men and women. Then, we performed multigroup structural equation modeling for men and women to verify the factors appropriate for men and women. The results demonstrated the model’s goodness of fit to be: χ^2^ = 226.274 (df = 34), *p* = 0.001, CFI = 0.944, AGFI = 0.926, RMSEA = 0.067, AIC = 270.274 (Model 1). Next, we analyzed the model without equality constraints for all other paths. The model’s goodness-of-fit was as follows: χ^2^ = 227.124 (df = 39), *p* = 0.001, CFI = 0.945, AGFI = 0.95, RMSEA = 0.062, AIC = 261.124 (Model 2). The goodness of fit for both models was sufficient to allow for the adoption of these models; however, we selected Model 2 because of its higher CFI and AGFI and lower AIC and RMSEA values. Two factors were appropriate for men and women. Factor 1 included four items representing a distrust of others (hereafter, “distrust of others”), while Factor 2 included three items representing the desire to solve one’s problems alone without help (hereafter, “solving problems alone”). The internal consistency coefficient for men and women was within an acceptable range (α = 0.63–0.85).

### 3.3. Gender Difference for Each Variable

We conducted a *t*-test using help-seeking intention, barriers to help-seeking, and FTP as the dependent variables to examine gender difference. The women had a higher help-seeking intention score (t_(1272)_ = 7.77, *p* < 0.01, d = 0.44) than that of the men (Table 4). For barriers to help-seeking, there were significant differences on “solving problems alone” (t_(1272)_ = 5.73, *p* < 0.01, d = 0.32) and “distrust of others” (t_(1272)_ = 7.02, *p* < 0.01, d = 0.39), with men scoring higher on both factors than women. A strong significant positive correlation was observed between “solving problems alone” and “distrust of others” (r = 0.56, *p* < 0.01).

For FTP, “resignation to the future” score was higher for women than for men (t_(1272)_ = 4.79, *p* < 0.01, d = 0.27). In contrast, for “future anxiety,” there was no significant difference between women and men (t_(1272)_ = 1.35, *p* = 0.18, d = 0.08). A moderate significant positive correlation was observed between “resignation to the future” and “future anxiety” (r = 0.33, *p* < 0.01).

### 3.4. Structural Equation Modeling Analysis

We performed multigroup structural equation modeling for men and women to verify the hypothesis model (Figure 1). Paths that were not significant for both men and women were removed, and each path for men and women was analyzed without equality constraints. The model’s goodness of fit was as follows: χ^2^ = 10.550 (df = 6), *p* = 0.103, CFI = 0.997, AGFI = 0.975, RMSEA = 0.024, AIC = 142.550 (Model 1). Next, we analyzed the model without equality constraints for paths for which significance differed between the two groups and with equality constraints for all other paths. The model’s goodness of fit was as follows: χ^2^ = 24.741 (df = 20), *p* = 0.212, CFI = 0.997, AGFI = 0.983, RMSEA = 0.014, AIC = 128.741 (Model 2). The goodness of fit for both models was sufficient to allow for the adoption of these models, but we selected Model 2 because of its higher AGFI and lower AIC and RMSEA values (Figure 2). The analysis showed significant negative paths for both men and women from current economic status to “resignation to the future,” “future anxiety,” and “distrust of others.” Furthermore, for women, significant negative paths were confirmed from current economic status to “solving problems alone” and from childhood economic status to “resignation to the future,” “future anxiety,” “distrust of others,” and “solving problems alone.” For both sexes, a significant negative path was observed from “distrust of others” to help-seeking intentions. In men, a significant negative path was also seen from “resignation to the future” to help-seeking intentions, along with a significant positive path from current economic status to help-seeking intentions. In women, a significant negative path was noted from “solving problems alone” to help-seeking intentions as well as a significant positive path from childhood economic status to help-seeking intentions.

## 4. Discussion

This study aimed to reveal the psychological mechanisms that work to suppress help-seeking behavior in middle-aged and older men and women who live alone. The objectives were (1) to verify whether economic status (a) correlates with barriers to help-seeking and perceived future time perspective and (b) suppresses intention to seek help and (2) to identify gender differences on all the measured variables.

The results revealed that help-seeking intention was lower in men than in women. This finding was consistent with those from previous studies [12,37]. Past research has also reported that women have more abundant social relationships than men [38,39]. During difficult times, these relationships may serve as a source of support and, therefore, facilitate help-seeking behavior. Regarding barriers to seeking help, men who lived alone were found to be more likely than women who lived alone to try to solve problems by themselves and to have a greater distrust of others. Past research has also indicated that such barriers to help-seeking are characteristic of men and are thought to be influenced by norms of masculinity, such as “a man should appear aggressive, and be physically and emotionally strong” [40]. In contrast, women are seen as more proactive help-seekers because dependence on others has been normalized as a “feminine” gender role [15]. Additionally, women have also been found to use more support-seeking as stress-coping strategies than men do [28,29]. “Resignation to the future” was higher among men than among women. Previous studies have shown that men have a lower FTP than women [26,41]. While women set diverse goals for their future, including goals related to work, family, and leisure, men tend to focus primarily on career-related matters. An interview survey conducted by Yetter [42] also demonstrated that older men who live alone perceive a solitary lifestyle as an opportunity to enjoy freedom and independence. These findings suggest that it is important that middle-aged and older men who live by themselves—a group characterized by difficulty looking to the future—be encouraged to begin planning for their future at an earlier stage, before older adulthood, to develop an FTP.

The results of this study revealed that there was no significant difference between men and women for “future anxiety,” even though current economic status was better for women than men in this study. In Japan, a high percentage of middle-aged and older women are not full-time workers, and the percentage of women in managerial positions remains low [43]. Men in their 50s make, on average, approximately double the amount women make (National Tax Administration Agency, 2019); their pension benefits are also higher than those of their women counterparts [44]. This disparity may be a factor contributing to the same level of anxiety regarding the future despite the current good economic status among women living alone.

The correlation analysis showed a significant positive correlation between “resignation to the future” and “future anxiety,” regardless of the participants’ sex. Emotional suppression, or the conscious inhibition of one’s own emotional expressive behavior while emotionally aroused, is seen as a strategy for avoiding events that one perceives to be undesirable [45]. While this behavior does not reduce negative emotions, it causes high levels of distress [46] and affects the individual’s mental and physical health [47,48]. Research on the use of avoidance as a coping method among middle-aged and older adults shows that avoidant coping is associated with increased stress response and depressive tendencies [28]. Similarly, “resignation to the future” can also be seen as a way to avoid or suppress anxiety about the future. This finding also has a clinical implication: minimizing anxiety about the future may contribute to better mental health among middle-aged and older men living alone [49]. Future research should explore the causal relationship between “resignation to the future” and “future anxiety” as well as the mechanism by which they influence mental and physical health.

Furthermore, the results of the multigroup structural equation modeling demonstrated that “resignation to the future,” “future anxiety,” and “distrust of others” were higher in both men and women who had a poorer economic status at present. Previous studies have also indicated an association between economic status and help-seeking behavior [50,51]. It was also shown that poorer economic status at present is associated with showing attitudes of “distrust of others,” which suppressed the intention to seek help among the participants, regardless of gender. This suggests that if a relationship of trust with the person that needs to be consulted is not established, it will not lead to a request for assistance in spite of their economic difficulties. Thus, it is essential to forge a trusting relationship with them to promote help-seeking behaviors. A difference was observed in this study between men and women with respect to the path leading to help-seeking intentions. For men living alone, the poorer their current economic status, the more they exhibited an attitude of “resignation to the future,” which in turn suppressed their intention to seek help. This finding supports the hypothesis proposed in this study; additionally, it suggests that the characteristic attitude of such men of giving up on hope for the future may suppress their help-seeking behavior. Previous work suggested that individuals who suppress the expression of emotions lose opportunities to build close relationships with others and fail to elicit social support [52]. Participants in our study were born before the 1960s and belonged to a generation that was strongly influenced by the gender role attitude of “men at work, women at home” [18]. In other words, not only do middle-aged and older men living alone have fewer sources of social support, but their experiences of economic difficulty also contradict the traditional masculine norms that emphasize “self-sufficiency.” This intensifies their feeling of resignation to the future, thereby inhibiting their help-seeking behavior.

In the case of middle-aged and older women who lived alone, the poorer their current and childhood economic statuses, the stronger their desire to solve problems by themselves, thereby resulting in suppression of help-seeking behavior. Through a retrospective study, Zahodne et al. [29] found that a bigger social support system in childhood influences the size of an individual’s social network in adulthood via educational attainment. According to a survey [5], the educational attainment for women was much lower than that for men if their family’s financial circumstances during childhood were challenging. In particular, the participants of this study would have graduated from school and began working before the Equal Employment Opportunity Act was enacted in 1985; thus, they belong to a generation in which women faced a significant disadvantage compared to men [53]. It has also been shown that older women living alone have a much higher risk of experiencing financial difficulties than men owing to low income and low or no pension [54]. Therefore, in addition to being raised in a harsh economic environment, the habit of solving problems by themselves—learned out of the need to solve an array of problems on their own while advancing through school and career in an environment in which women were extremely disadvantaged—may be a factor that suppresses help-seeking behavior in middle-aged and older women.

Thus, policies and programs that aim to promote help-seeking among middle-aged and older adults living alone who face difficulties in life need to consider gender differences in psychological factors that suppress help-seeking. For men living alone, it may be important to build trusting relationships with consultants and to promote their future perspective. For women, it may be important to build trusting relationships with consultants and to encourage them to consider whether their problems are about relying on people or solving them by themselves. In the future, it will be necessary to use qualitative methods to understand middle-aged and older adults’ perceptions of these psychological factors that suppress help-seeking behavior.

However, the findings of this study need to be interpreted in light of the following limitations. First, since this was a cross-sectional study, the causal relationships between help-seeking intentions and other variables remain unclear. This should be explored in the future using a longitudinal research design. Furthermore, since the survey was conducted via postal mail and those who live alone are less likely to respond to surveys [32], this study’s results should be interpreted very carefully. Second, our results are based on a sample from Japan and thus may not be generalizable to other countries or cultures. Therefore, it is necessary to collect additional data from various countries and age groups. Third, this study included family members, friends, and acquaintances as sources to seek help from but did not address other potential options such as professionals or public institutions. In the future, it will be necessary to study different avenues for help-seeking and the challenges associated with them for middle-aged and older adults living alone.

Despite these limitations, this study was able to clarify the psychosocial mechanisms that suppress help-seeking behavior among middle-aged and older adults living alone, which is a cohort that has not been studied sufficiently in the past. This study offers useful recommendations for policy and research to prevent indigence and social isolation. Specifically, policies and programs promoting help-seeking among middle-aged and older adults living alone and facing life difficulties should consider gender differences in psychological factors that suppress help-seeking.

## 5. Conclusions

The present study shows that at present, lower economic status is associated with attitudes of “distrust of others,” which suppressed the intention to seek help among the middle-aged and older adults living alone, regardless of gender. A difference was observed in this study between men and women with respect to the path leading to help-seeking intentions. For men, the lower their current economic status, the more they exhibited an attitude of “resignation to the future,” which in turn suppressed their intention to seek help. For women, the lower their current and childhood economic statuses, the stronger their desire to solve problems by themselves, thereby resulting in suppression of help-seeking behavior. Therefore, policies and programs that aim to promote help-seeking among middle-aged and older adults living alone, who face difficulties in life, need to consider gender differences in psychological factors that suppress help-seeking.

## Figures and Tables

**Figure 1 ijerph-19-10620-f001:**
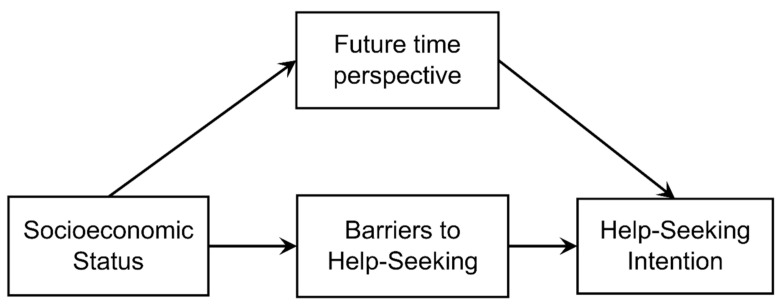
Hypothesis model for help-seeking behavior among middle-aged and older men.

**Figure 2 ijerph-19-10620-f002:**
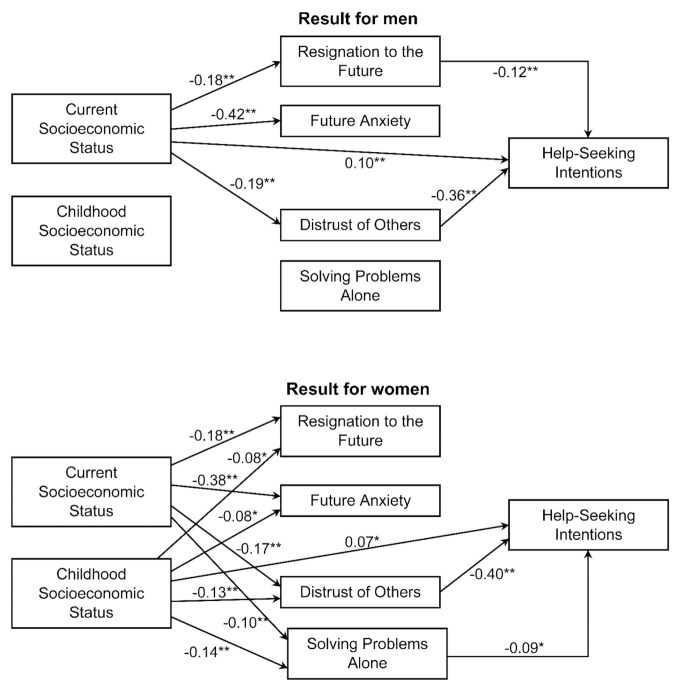
Multigroup structural equation modeling results for men and women. All values are standardized estimates. Age, error terms, and covariance have been omitted. Only paths significant at the 5% level are shown. (* *p* < 0.05, ** *p* < 0.01).

**Table 1 ijerph-19-10620-t001:** Participant Characteristics.

	*n* (%) or M (SD)	
Age	63.97	±8.63
Sex		
Male	658	(51.6)
Female	616	(48.4)
Current economic status (1–5)	2.90	±0.97
Childhood economic status (1–5)	2.90	±0.97

Note. Current and childhood economic status were reported on a scale of 1 to 5, where 1 = extremely difficult and 5 = extremely comfortable.

**Table 2 ijerph-19-10620-t002:** Results of the Factor Analysis for The Revised Perceived Future Time Perspective Scale (Principal Factor Method, Promax rotation).

Factor and Items		Factor Loading	α	% Variance
Factor 1: Future Anxiety			0.73	38.21
	6. I get depressed when I think about the future	0.95		
	8. I feel bleak when I think about what comes next	0.87		
	4. I get anxious when I think about life in the future	0.77		
Factor 2: Resignation to the future			0.89	18.53
	2. I have no interest in my future	0.80		
	5. I do not care what happens to me	0.65		
	1. I am leaving my future up to chance	0.56		
	7. Thinking about the future is a hassle	0.51		

Note. Item 3 (“Things will work out somehow”) in the scale was excluded during the factor analysis.

**Table 3 ijerph-19-10620-t003:** Results of the Factor Analysis for the Barriers to Help-Seeking Scale (Principal Factor Method, Promax rotation).

Factors and Items		Factor Loading	α	% Variance
Factor 1: Distrust of Others			0.85	46.24
	7. Even if I ask for help, they do not understand	0.94		
	10. There is nobody I trust and can ask for help	0.73		
	6. I do not want to talk about my feelings with someone else	0.68		
	5. Even if I ask for help, things will not be resolved	0.68		
Factor 2: Solving Problems Alone			0.67	7.53
	1. I want to solve my problems by myself	0.70		
	4. I do not want to rely on others	0.70		
	8. I do not want to trouble anyone	0.48		

Note. Items 2 (“I do not want to tell my secrets to other people”), 3 (“It is embarrassing to ask for help”), and 9 (“I do not want to be thought of as someone who is less capable”) in the scale were excluded during the factor analysis.

**Table 4 ijerph-19-10620-t004:** Results of the *t*-test.

Variable	Overall	Men	Women	*p*-Value
	M (SD)	M (SD)	M (SD)	
Help-Seeking Intentions (0–35)	10.90 (6.26)	9.62 (6.43)	12.28 (5.77)	0.001
Barriers to Help-Seeking				
Solving Problems Alone (3–15)	11.78 (2.45)	12.16 (2.33)	11.38 (2.52)	0.001
Distrust of Others (4–20)	12.38 (3.93)	13.12 (3.86)	11.60 (3.85)	0.001
Perceived Future Time Perspective				
Future Anxiety (3–12)	7.55 (2.62)	7.64 (2.58)	7.44 (2.67)	0.178
Resignation to the Future (4–16)	8.76 (2.68)	9.11 (2.79)	8.39 (2.50)	0.001

Note. *p*-values obtained using *t*-test.

## Data Availability

The data presented in this study are available on request from the corresponding author. The data are not publicly available due to privacy/ethical concerns.

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
