# Peer review of "Psychological Factors That Suppress Help-Seeking among Middle-Aged and Older Adults Living Alone"

_ijerph, 2022, doi:10.3390/ijerph191710620_

Round 1

Reviewer 1 Report

Dear Authors,

congratulations on your excellent work and for choosing such an important topic. On a global scale, we are facing with ageing population and the persons who live alone, and the topic you choose to investigate is so underestimated in most of the countries, even in the country from I come from. Hope the results from your paper will reach wide range of stakeholders, institutions, science and general public population.

I have a few recommendations for your paper, as follow:

The topic is good and points to a problem that has been neglected in the community and is getting bigger.

The title reflects the content of the paper.

The abstract clearly describes the problem and the results of the research. I would recommend that the last two sentences from the Abstract be moved to the end of the paper as recommendations.

The introduction is well written, supported by current references and introduces the reader to the issue. I think that the Introduction is too long and that it would be necessary to shorten and concretize it a bit. The section describing Suppress Help-Seeking in middle-aged and older men who live alone is too broad and becomes tedious for the reader.

Materials and Methods are well arranged and described. Given that the authors wrote that the fully completed document returned to the authors was considered the consent of the participants to the research, it would be good to state one sentence whether the participants received a written description of the research and instructions for completing all questionnaires.

The results are well presented.

The discussion is well written with a comparison with other research, but it would be good to highlight the clarification of both hypotheses more clearly (now only one is very clearly highlighted).

Good luck and keep going on your good work.

Kind regards

Author Response

Dear Reviewer 1:

We thank you for your appreciation of parts of our manuscript, and we hope to provide a clear picture through our revisions based on your suggestions. The revised text in the manuscript is indicated by track changes, which denotes places of correction. We have used underlined red font to indicate new or revised text in the manuscript.

  1. I would recommend that the last two sentences from the Abstract be moved to the end of the paper as recommendations.

Response: We have moved the last two sentences from the Abstract to the end of the paper as per your recommendation (p. 10, new line 407-413):

Despite these limitations, this study was able to clarify the psychosocial mechanisms that suppress help-seeking behavior among middle-aged and older adults living alone, a cohort that has not been studied sufficiently in the past. This study offers useful recommendations for policy and research to prevent indigence and social isolation. Specifically, policies and programs promoting help-seeking among middle-aged and older adults living alone and facing life difficulties should consider gender differences in psychological factors that suppress help-seeking.

According to above correction, we have added one reference as follows:

  1. Hanibuchi T,; Nakaya T,; Murakami A,; Hanaoka K. Regional Differences in Survey Response Rates and Their Individual and Geographic Determinants: A Multilevel Analysis. Geogr. Rev. Jpn. series A. 2012, 85 (5), 447-467. doi.org/10.4157/grj.85.447

  1. I think that the Introduction is too long and that it would be necessary to shorten and concretize it a bit. The section describing Suppress Help-Seeking in middle-aged and older men who live alone is too broad and becomes tedious for the reader.

Response: We have shortened and concretized the Introduction section by removing the description of suppressed help-seeking in middle-aged and older men who live alone (p. 2, new line 52-82).

Studies have also found that men are less likely than women to seek help for medical problems, substance abuse, and psychological distress [10–13]. (p. 2, new line 52-54). Nevertheless, few studies explain why middle-aged and older men living alone tend to avoid seeking help despite difficulties. Therefore, this study seeks to identify the psychological factors that suppress help-seeking in this population.

They discovered that these men “were not thinking about the future despite facing severe difficulties in everyday life. (p. 2, lines 74-75). Moreover, an online survey of men and women between the ages of 50 and 79 found that men who lived by themselves were more likely to give up hope for the future compared to men who lived with others, women who lived alone, and women who lived with others [23].

FTP is also strongly correlated with planning for life after retirement; that is, individuals who are high on FTP tend to be more involved in planning for post-retirement life [26,27]. (p. 2, lines 80-82) Furthermore, in a longitudinal study, Korff and Biemann [28] found that subjective health, loneliness, and life satisfaction influence FTP.

  1. Given that the authors wrote that the fully completed document returned to the authors was considered the consent of the participants to the research, it would be good to state one sentence whether the participants received a written description of the research and instructions for completing all questionnaires.

Response: We have added a sentence explaining that the participants received a written description of the research and instructions for completing all questionnaires as below. (p.3, new line 116-119):

This study was approved by the Research Ethics Board of XX (approval number: XX; approved on August 4, 2020). The participants received a written description of the research and instructions for completing all questionnaires.

  1. It would be good to highlight the clarification of both hypotheses more clearly (now only one is very clearly highlighted).

Response: We have highlighted the clarification of the other hypothesis as below (p. 2, new line 85-89):

However, women are seen as more proactive help-seekers because dependence on others has been normalized as a “feminine” gender role [15]. Additionally, a previous study also showed that women use more coping strategies than men do [28,29]. In other words, it is possible that women are more proactive problem-solvers than men are and are therefore more open to the idea of seeking help.

Reviewer 2 Report

The authors should describe the sampling strategy, the response rate (currently which has to be calculated by the readers), the explanation for the low response rate, and the comparison of individual characteristics between the sample and prior studies. Without such information, readers would not be able to tell if the results of the study are reliable. 

Author Response

Dear Reviewer 2:

We thank you for your appreciation of parts of our manuscript, and we hope to provide a clear picture through our revisions based on your suggestions. The revised text in the manuscript is indicated by track changes, which denotes places of correction. We have used underlined red font to indicate new or revised text in the manuscript.

  1. The authors should describe the sampling strategy, the response rate (currently which has to be calculated by the readers), the explanation for the low response rate, and the comparison of individual characteristics between the sample and prior studies. Without such information, readers would not be able to tell if the results of the study are reliable. 

Response: We have added an explanation for the low response rate, and the comparison of individual characteristics, including a new reference (p.3 new line 108-113):

Using a random sampling method, 4,000 residents were selected from 247,445 residents between the ages of 50 and 79 who were listed in the Basic Resident Register of Tokyo’s Ward A (as of January 2020) and who lived alone (response rate: 45.7% [n = 1,829]); a postal survey was conducted in December 2020. The response rates from single-person households, urban areas, and working generations were relatively low [32], which reflects the individual characteristics in this study.

Reviewer 3 Report

2.2. Measures. It would be interesting to add data on the quality of the instruments used (e.g. reliability, validity).

2.2.2. Psychological Barriers to Help-seeking. It is not clear to me whether 10 questions are used or only 1 (the one that is written in the text). More explanation is needed. 

2.2.3. Help-Seeking Intention. Same comment as above, is only one question from this scale used?

Table 1. Participant Characteristics. Same M and SD Current and Childhood economic status  (2,90  and ± 0.97)? Is it right?

Author Response

Dear Reviewer 3:

We thank you for your appreciation of parts of our manuscript, and we hope to provide a clear picture through our revisions based on your suggestions. The revised text in the manuscript is indicated by track changes, which denotes places of correction. We have used underlined red font to indicate new or revised text in the manuscript.

  1. 2.2. Measures. It would be interesting to add data on the quality of the instruments used (e.g. reliability, validity).

Response: Thank you. We have described the quality of the instruments used (reliability and validity) in “3.2 Scale Structure” (p.5 new line 198-219):   

  1. 2.2.2. Psychological Barriers to Help-seeking. It is not clear to me whether 10 questions are used or only 1 (the one that is written in the text). More explanation is needed. 

Response: I added more detail on the scale of Psychological Barriers to Help-seeking as follows (p.3 new line 138-141):

In this study, participants were asked “What do you think about asking someone (e.g., a relative, friend, or acquaintance) for help?” on a five-point scale (5 = “strongly agree” to 1 = “strongly disagree”). We used all 10 questions to measure psychological barriers to help-seeking.

  1. 2.2.3. Help-Seeking Intention. Same comment as above, is only one question from this scale used?

Response: I added more detail on the scale of Help-Seeking Intention as follows (p.4 new line 151-157):

The following helper categories were selected based on the results of a preliminary survey [23], as well as the attributes of the participants of this study (middle-aged and older adults living alone): 1) intimate partner, 2) friend, 3) colleague from work (including a former workplace), 4) parent, 5) sibling, 6) son/daughter, and 7) neighbor. The total score for each item was calculated and used as the help-seeking intention score. The score range is 0–35 points; the higher the score, the greater the intention of help-seeking.

  1. Table 1. Participant Characteristics. Same M and SD Current and Childhood economic status  (2,90  and ± 0.97)? Is it right?

Response: Thank you. We have checked again and confirmed that current and childhood economic status had almost identical M and SD as below. As the numbers beyond the second decimal point were suppressed in this study, the M and SD displayed in Table 1 became identical.

  • Current economic status: M=2.895, SD=0.973
  • Childhood economic status: M=2.898, SD=0.970

Round 2

Reviewer 2 Report

Thank you for improving the manuscript by adding a few sentences to explain how the samples were selected and the ratio of the response rate. However, I do not think such an explanation would be sufficient. As you mentioned in the response letter, the survey was conducted via postal mail and those who lived alone were less likely to respond to the survey. Therefore, the results of this study should be interpreted very carefully. Furthermore, the authors should clearly address this limitation in the discussion section. 

Author Response

Response to Reviewer 2

Dear Reviewer 2:

We thank you for your appreciation of parts of our manuscript, and we hope that our revisions, based on your suggestions, help enhance clarity. The tracked changes in the manuscript indicate the parts of the manuscript that have been revised. Additionally, the red-colored texts indicate the new and revised text in the manuscript.

  1. I do not think such an explanation would be sufficient. As you mentioned in the response letter, the survey was conducted via postal mail and those who lived alone were less likely to respond to the survey. Therefore, the results of this study should be interpreted very carefully. Furthermore, the authors should clearly address this limitation in the discussion section.

Response: We have added the fact that those who live alone are less likely to respond to the survey as a limitation of this study (p.10, new lines 378–380):

However, the findings of this study need to be interpreted in light of the following limitations. First, since this was a cross-sectional study, the causal relationships between help-seeking intentions and other variables remain unclear. This should be explored in the future using a longitudinal research design. Furthermore, since the survey was conducted via postal mail and those who live alone are less likely to respond to surveys [32], this study’s results should be interpreted very carefully.
